# Lipids and Transaminase in Antiretroviral-Treatment-Experienced People Living with HIV, Switching to a Doravirine-Based vs. a Rilpivirine-Based Regimen: Data from a Real-Life Setting

**DOI:** 10.3390/v15071612

**Published:** 2023-07-23

**Authors:** Paolo Maggi, Elena Delfina Ricci, Canio Vito Martinelli, Giuseppe Vittorio De Socio, Nicola Squillace, Chiara Molteni, Addolorata Masiello, Giancarlo Orofino, Barbara Menzaghi, Rita Bellagamba, Francesca Vichi, Benedetto Maurizio Celesia, Giordano Madeddu, Giovanni Francesco Pellicanò, Maria Aurora Carleo, Antonio Cascio, Andrea Parisini, Lucia Taramasso, Laura Valsecchi, Leonardo Calza, Stefano Rusconi, Eleonora Sarchi, Salvatore Martini, Olivia Bargiacchi, Katia Falasca, Giovanni Cenderello, Sergio Ferrara, Antonio Di Biagio, Paolo Bonfanti

**Affiliations:** 1Infectious Diseases Unit, AORN Sant’Anna e San Sebastiano, 81100 Caserta, Italy; paolo.maggi@unicampania.it (P.M.); dora.80@live.it (A.M.); 2Fondazione ASIA Onlus, 20090 Buccinasco, Italy; 3AOU Infectious and Tropical Diseases, Careggi Hospital, 50134 Florence, Italy; martinellica@aou-careggi.toscana.it; 4Unit of Infectious Diseases, Santa Maria Hospital,06156 Perugia, Italy; giuseppe.desocio@ospedale.perugia.it; 5Infectious Diseases Unit, Fondazione IRCCS San Gerardo dei Tintori, 20900 Monza, Italy; nicola.squillace@irccs-sangerardo.it (N.S.); paolo.bonfanti@unimib.it (P.B.); 6Unit of Infectious Diseases, A. Manzoni Hospital, 23900 Lecco, Italy; c.molteni@asst-lecco.it; 7Division I of Infectious and Tropical Diseases, ASL Città di Torino, 10149 Torino, Italy; giancarlo.orofino@aslcittaditorino.it; 8Unit of Infectious Diseases, ASST della Valle Olona, 21052 Busto Arsizio, Italy; barbara.menzaghi@asst-valleolona.it; 9National Institute for Infectious Diseases Lazzaro Spallanzani Institute for Hospitalization and Care Scientific, Lazio, 00161 Roma, Italy; rita.bellagamba@inmi.it; 10SOC 1 USLCENTRO Firenze, Unit of Infectious Diseases, Santa Maria Annunziata Hospital, 50012 Florence, Italy; francesca.vichi@uslcentro.toscana.it; 11Unit of Infectious Diseases, Garibaldi Hospital, 95124 Catania, Italy; bmcelesia@gmail.com; 12Unit of Infectious Diseases, Department of Medicine, Surgery and Pharmacy, University of Sassari, 07100 Sassari, Italy; giordano@uniss.it; 13Unit of Infectious Diseases, Department of Human Pathology of the Adult and the Developmental Age ‘G. Barresi’, University of Messina, 98125 Messina, Italy; gpellicano@unime.it; 14Infectious Diseases and Gender Medicine Unit, Cotugno Hospital, AO dei Colli, 80131 Naples, Italy; mariaaurora.carleo@ospedalideicolli.it; 15Unit of Infectious Diseases, Department of Health Promotion, Mother and Child Care, Internal Medicine and Medical Specialties, University of Palermo, 90127 Palermo, Italy; antonio.cascio03@unipa.it; 16Department of Infectious Diseases, Galliera Hospital, 16128 Genoa, Italy; andrea.parisini@galliera.it; 17Clinic of Infectious Diseases, IRCCS Policlinico San Martino Hospital, University of Genoa, 16132 Genoa, Italy; taramasso.lucia@gmail.com (L.T.); antonio.dibiagio@hsanmartino.it (A.D.B.); 181st Department of Infectious Diseases, ASST Fatebenefratelli Sacco, 20157 Milan, Italy; laura.valsecchi@asst-fbf-sacco.it; 19Infectious Diseases Unit, IRCCS Policlinico Sant’ Orsola, Department of Medical Surgical Science, University of Bologna, 40138 Bologna, Italy; leonardo.calza@unibo.it; 20Infectious Diseases Unit, Ospedale Civile di Legnano, ASST Ovest Milanese, and DIBIC Luigi Sacco, Università degli Studi di Milano, 20025 Legnano, Italy; stefano.rusconi@unimi.it; 21Infectious Diseases Unit, Santi Antonio e Biagio e Cesare Arrigo Hospital, 15121 Alessandria, Italy; eleonora.sarchi@ospedale.al.it; 22Infectious Disease Unit, University Hospital Luigi Vanvitelli, 80138 Naples, Italy; salvatoremartini76@gmail.com; 23Unit of Infectious Diseases, Ospedale Maggiore della Carità, 28100 Novara, Italy; olivia.bargiacchi@gmail.com; 24Clinic of Infectious Diseases, Department of Medicine and Science of Aging, G. D’Annunzio University, Chieti-Pescara, 66100 Chieti, Italy; k.falasca@unich.it; 25Department of Infectious Diseases, Sanremo Hospital, 18038 Sanremo, Italy; g.cenderello@asl1.liguria.it; 26Unit of Infectious Diseases, Department of Clinical and Experimental Medicine, University of Foggia, 71122 Foggia, Italy; sferrara@ospedaliriunitifoggia.it; 27School of Surgery and Medicine, University of Milano-Bicocca, 20126 Milan, Italy

**Keywords:** HIV infection, ART-experienced, doravirine, rilpivirine, adverse events, metabolic safety, hepatic safety

## Abstract

Doravirine (DOR) is a newly approved non-nucleoside reverse transcriptase inhibitor (NNRTI). We aimed to investigate, in a real-life setting, how switching to a DOR-based regimen rather than a rilpivirine (RPV)-based regimen impacted metabolic and hepatic safety. The analysis included 551 antiretroviral treatment (ART)-experienced people living with HIV (PLWH), starting RPV-based or DOR-based regimens with viral load < 200 copies/mL, baseline (T0), and at least one control visit (6-month visit, T1). We enrolled 295 PLWH in the RPV and 256 in the DOR cohort. At T1, total cholesterol (TC), low-density lipoprotein-C (LDL-C), and triglycerides significantly decreased in both DOR and RPV cohorts, while high-density lipoprotein-C (HDL-C) only decreased in RPV-treated people. Consistently, the TC/HDL-C ratio declined more markedly in the DOR (−0.36, *p* < 0.0001) than in the RPV cohort (−0.08, *p* = 0.25) (comparison *p* = 0.39). Similar trends were observed when excluding the PLWH on lipid-lowering treatment from the analysis. People with normal alanine aminotransferase (ALT) levels showed a slight ALT increase in both cohorts, and those with baseline ALT > 40 IU/L experienced a significant decline (−14 IU/L, *p* = 0.008) only in the DOR cohort. Lipid profile improved in both cohorts, and there was a significant reduction in ALT in PLWH with higher-than-normal baseline levels on DOR-based ART.

## 1. Introduction

HIV infection has now moved from a life-threatening condition to a chronic, well-controlled one. However, people living with HIV (PLWH) are still more prone to age-related diseases, especially atherosclerosis [1]. In this, an important role is played by the residual inflammation [2] and suboptimal HIV suppression [3]. However, the possible effect of antiretroviral therapy (ART) in weight gain [4] and in changing the lipid profile, especially for older non-nucleoside reverse transcriptase inhibitors (NNRTIs) [5], could still play a role.

Doravirine (DOR) is a newly approved antiretroviral belonging to the class of NNRTI. This agent seems well tolerated in both ART-naïve and ART-experienced PLWH—in the phase 3 DRIVE-FORWARD trial, DOR demonstrated a superior lipid profile compared with darunavir-ritonavir after 48 weeks of combination treatment with two nucleoside reverse-transcriptase inhibitors (NRTIs) in antiretroviral-naïve adults [6].

In the phase 3 DRIVE-AHEAD trial, DOR, in combination with lamivudine (3TC) and tenofovir disoproxil fumarate (TDF), showed minimal changes in low-density lipoprotein cholesterol (LDL-C) and non-high-density lipoprotein cholesterol (non-HDL-C) at week 48 compared with efavirenz (EFV), emtricitabine (FTC), and TDF [7].

The phase 3 DRIVE-SHIFT trial evaluated switching from a stable antiretroviral regimen to once-daily DOR/3TC/TDF in adults with HIV-1 suppressed for ≥6 months and no previous virologic failure. Reductions in fasting lipids were observed at 24 weeks after the switch and maintained through to week 144. The mean weight change from the switch to week 144 was +1.4 kg in the immediate switch group and +1.2 kg in the delayed switch group [8].

Previous real-life data also confirm the improved profile of lipid metabolism of PLWH switching to DOR-based regimens [9].

Data on the hepatic safety report that blood testing may commonly reveal increased alanine aminotransferase (ALT) during treatment with DOR [10], whereas aspartate aminotransferase (AST) increase is rare, but results from real life are scant.

Our aim was to investigate the effect on metabolic and hepatic safety of switching to a DOR-based regimen in a real-life setting. We compared people on a DOR regime with a cohort of PLWH switching to rilpivirine (RPV)-based regimens, as DOR and RPV are the two second-generation NNRTIs widely used in modern ART.

## 2. Materials and Methods

We analyzed data from the SCOLTA (Surveillance Cohort Long-Term Toxicity Antiretrovirals) prospective database. The SCOLTA project is a multicenter observational study, started in 2002, following prospective PLWH who start to take new antiretroviral drugs, to identify toxicities and adverse events (AEs) in a real-life setting [11]. The SCOLTA project uses an online pharmacovigilance program and involves 25 Italian Infectious Disease Centers (www.cisai.it).

Briefly, both ART-naïve and ART-experienced PLWH can be included in SCOLTA, if they are >18 years of age and agree to study entry. Clinical data include sex, age, ethnicity, weight, height, CDC stage, and previous ART history. Laboratory data include HIV-RNA, CD4 +T cell count, and biochemical data, and are prospectively collected in anonymous form in a central database every six months. AEs are collected prospectively as soon as they are clinically observed.

The RPV cohort was opened to enrolment between January 2013 and September 2017, when RPV could no longer be considered a “newly marketed drug”. The first participant was enrolled in the DOR cohort in February 2020 and the enrolment is still ongoing.

In this analysis, we aimed to compare the metabolic safety of DOR and RPV. To this end, we included all ART-experienced PLWH with available data for at least total cholesterol or ALT at baseline (T0) and 6-month follow-up (T1).

The original study protocol was approved on 18 September 2002, and three amendments were approved on 13 June 2013, 20 December 2019, and 12 May 2020 by the coordinating center at Hospital “L. Sacco”-University of Milan, Milan (Italy) and thereafter by all participating centers. Written consent for study participation was obtained from all participants, and the study was conducted in accordance with the ethical standards laid down in the 1964 Declaration of Helsinki and its later amendments and by Italian national laws.

### Statistical Analysis

Data were described using mean and standard deviation (SD) for normally distributed continuous variables, median and interquartile range (IQR) for non-normally distributed continuous variables, and frequency (%) for categorical and ordinal variables. Distribution normality was assessed using the graphical quantile–quantile method. Baseline differences between means were tested using the analysis of variance and between medians using the non-parametric Mann–Whitney test. Proportion comparisons were performed using the chi-square test.

Intragroup change from baseline was evaluated using the paired *t*-test. Changes from baseline between the two cohorts were compared using a multivariable general linear model, including variables that were significantly different between groups, at baseline, and potentially confounding (variables reported in the footnotes).

Frequencies of discontinuation for any reasons and adverse events during the first year of treatment were compared using the Kaplan–Meyer survival curves (and log-rank test). Reasons for interruptions were described.

All *p*-levels were two-sided, at the significance level < 0.05. All statistical analyses were performed using SAS for Windows 9.4 (SAS Institute, Cary, NC, USA).

## 3. Results

In this analysis, we included 295 people on RPV-based regimens (mostly on FTC/TDF/RPV, 90.2%) and 256 on DOR-based regimens (55.5% on 3TC/TDF/DOR).

Since the RPV cohort opened in January 2013 and closed in September 2017, whereas the DOR cohort opened in February 2020 and is currently ongoing, the median observation time was different, being 16 months (interquartile range 8–23) and 12 months (IQR 8–20), respectively, (*p* = 0.01). Thus, we limited the prospective analysis to the first year of study.

At baseline (Table 1), PLWH on DOR were older, had a higher body mass index (BMI), and had higher CD4+ cells/mm^3^. They were also more likely to be on treatment with lipid-lowering drugs (LLDs), although blood lipid levels and the total cholesterol (TC)/high-density lipoprotein-C (HDL-C) ratio were similar between groups. Despite similar rates of hepatic virus coinfections, liver enzymes, as well as the prevalence of people with values above the upper limit of normality (>40 IU/L) at T0, were lower in the DOR cohort than in the RPV cohort.

We also compared the baseline characteristics of PLWH in the DOR cohort according to their regimen (Appendix A): 3TC/TDF/DOR (DOR1, n = 142) or other DOR-based regimens (DOR2, n = 114). At baseline, people on DOR1 were younger (48.8 vs. 55.8 years, *p* < 0.0001) and had a lower frequency of hepatitis C virus (HCV) coinfection (9.2% vs. 27.2%, *p* = 0.0001). Their previous regimen included NNRTI more frequently (47.9% vs. 34.2%, *p* = 0.03) and, less frequently than DOR2, protease inhibitors (PIs) (13.4% vs. 39.5%, *p* < 0.0001), and integrase inhibitors (INI) (37.3% vs. 64.0%, *p* < 0.0001) (Appendix A). The two groups did not differ in terms of TC and HDL-C (Appendix A); however, the ratio between these variables was significantly higher in DOR1 than in DOR2 (4.2, IQR 3.4–5.1 vs. 3.8, IQR 3.3–4.6; *p* = 0.04) (Appendix A). Considering only participants on LLDs, no difference was observed (Appendix A). Despite the higher proportion of hepatic virus coinfections in the DOR2 group, AST and ALT were similar at baseline (Appendix A).

Where information was available, in this sample of PLWH with baseline HIVRNA < 200 copies/mL, 5.6% and 6.4% had detectable HIV RNA at T1 and 6.2% and 5.6% at T2 in the DOR and RPV cohorts, respectively.

Changes from the baseline are shown in Table 2, which also reports on the multivariable comparisons between groups.

BMI and weight change were minimal, although significantly higher than zero in the RPV-treated group. TC, low-density lipoprotein cholesterol (LDL-C), and triglycerides significantly decreased in both DOR and RPV, while HDL-C decreased in the RPV cohort but remained stable in the DOR cohort. Consistently, the TC/HDL-C ratio declined in the DOR cohort (−0.36, *p* < 0.0001) more markedly than in the RPV cohort (−0.08, *p* = not significant). The multivariate comparison, including baseline age, weight, previous use of PI and INI, previous and current use of tenofovir alafenamide (TAF) and TDF, and LLD use, did not show a significant difference between DOR- and RPV-based regimens. Similarly, LDL-C showed a reduction in both groups, more pronounced in the DOR group, but the difference was not statistically significant.

Similar results were observed when excluding the participants on LLD treatment at baseline from the analysis (Table 2).

Analyzing PLWH in DOR1 and DOR2 groups (Table 3), we found a similar improved lipid profile, irrespective of the type of regimen, after taking into account baseline age, weight, previous use of PI and INI, previous and current use of TAF and TDF, and LLD use.

Liver-enzyme changes were separately analyzed in people with baseline normal or abnormal levels. AST and ALT showed a statistically significant modification in both DOR and RPV, in people with levels ≤ 40 IU/L at baseline (Table 2).

On the contrary, we observed no significant modification in PLWH with AST > 40 IU/L at baseline, and a decline in those with baseline ALT > 40 IU/L in the DOR cohort (−14 IU/L, *p* = 0.008), whereas we observed no significant change in the RPV cohort (Table 2). The ALT decline was not statistically different in DOR1 and DOR2 groups (Table 3).

At baseline, 54 (23.3%) PLWH on DOR and 80 (27.3%) on RPV had a positive HBsAg and/or HCV test (*p* = 0.29). Comparing the baseline values of AST and ALT in strata of hepatic virus coinfection (Table 4), we found a significant difference between the DOR and the RPV groups with respect to ALT in HCV/HBsAg-negative, and in both AST and ALT in HCV/HBsAg-positive people. In negative PLWH on RPV-based regimens, both AST and ALT increased significantly over the first 6 months of treatment, whereas no significant variation was observed in positive PLWH. ALT change from baseline was significantly higher in the RPV group for hepatic-virus-negative persons. Including age, previous and current use of TDF and TAF, baseline level of the variable, hepatic virus coinfection status, and treatment group, we found no difference in AST with respect to coinfection status, whereas the adjusted mean AST variation was 0.6 IU/dL (95% CI −2.6, 3.8) in the DOR group and 4.1 IU/dL (95% CI 0.5, 7.8) for RPV-treated people (*p* = 0.02). A similar result emerged in ALT change from baseline, with the adjusted means being −1.0 IU/dL (95% CI −6.7, 4.6) and 7.6 IU/dL (95% CI 1.2, 14.1), respectively, (*p* = 0.002). Considering only PLWH with hepatic coinfections, those with detectable HCV-RNA (n = 50) had higher AST (46 vs. 23, *p* < 0.0001) and ALT (52 vs. 26, *p* < 0.0001) baseline levels than those with undetectable HCV-RNA (n = 84), but mean changes from baseline were not significantly different (AST: 1 IU/dl, 95% CI −7, 9 and 1 IU/dl, 95% CI −1, 4; ALT: 0 IU/dL, 95% CI −11, 12, and 4 IU/dL, 95% CI −1, 9, respectively), although these finding might have been due to the small sample size.

Performing the previous analyses in the subgroup of male PLWH, we found similar results (Appendix A).

Overall, 2 PLWH on DOR (1 with ALT > 40 IU/L at baseline) experienced a grade 2 ALT event, not leading to treatment interruption, whereas 20 on RPV (6 with baseline ALT ≤40 IU/L) had a grade 2 event (1 interrupted the treatment and recovered) and 3 a grade 3 event (baseline ALT > 40 IU/L).

Interruptions over the first year were 30 in the RPV and 9 in the DOR cohort: details are given in Table 5.

The treatment interruptions due to any reasons were different between treatment groups (log-rank *p* = 0.004, Figure 1a), whereas interruptions due to AEs were similar (log-rank *p* = 0.40, Figure 1b). A comparison of DOR1 and DOR2 study participants did not show any difference in treatment interruptions (4.2% vs. 2.6%, respectively, log-rank *p* = 0.53).

## 4. Discussion

In this study, we found high tolerability of DOR in a real-life context of ART-experienced people, who switched to DOR-containing regimens with levels of baseline HIV RNA <200 copies/mm^3^.

To our knowledge, this is the largest prospective observational study available to date on DOR use in ART-experienced PLWH, and the first study investigating the metabolic modifications (both on blood lipids and weight) in comparison with other regimens based on the only other second-generation NNRTI still used in modern ART, namely RPV.

The characteristics of the study participants at the baseline in the two cohorts showed some differences, with PLWH in the DOR cohort being older, with a higher BMI and body weight, and more frequently on treatment with LLDs. These differences could mirror the choice of the clinicians who seemed to consider DOR a particularly safe drug in people at risk for cardiovascular (CV) disease, in light of the data regarding lipid profile and weight gain from registration trials [6,7,8]. Not unexpectedly, DOR in association with TDF was more frequently prescribed in younger PLWH, at lower risk of renal and bone diseases as compared to older ones. The lower frequency of HCV coinfection is an evident consequence of the younger age of these PLWH. Several of them had a previous regimen based on an anchor drug belonging to the same class (NNRTI), but with a different backbone; also in this case, the shift to DOR in association with TDF could have been due to the fact that these two drugs are perceived as CV-friendly.

At follow-up, participants in DOR and RPV cohorts showed favorable modifications in blood lipids, even when limiting the observation to those not on LLDs. In particular, both regimens were associated with a significant reduction of TC, LDL-C, and triglycerides, although HDL-C also decreased in the RPV group. DOR also seemed more efficient than RPV in reducing the TC/HDL-C ratio, both in the whole sample and in the subgroup of those not on LLDs, although the difference was not significant.

When we considered the differences between DOR1 and DOR2 groups, the first regimen was associated with a more marked decrease in LDL-C and TC/HDL-C ratio, although no statistical significance was retrieved; this could have been due to the statin-like effect of TDF [12]. Comparable results emerged in a real-life study from Mazzitelli et al. [13], who reported a significant reduction in lipids (both cholesterol and triglycerides) in 52 PLWH, at 24 weeks of observation after initiating a DOR-based regimen. Noteworthy, in both groups, weight and BMI remained substantially unchanged, suggesting the safety of the two drugs in terms of weight-gain risk.

We gave particular attention to the changes in liver enzymes being the NNRTI class at risk for AST and/or ALT elevation [14].

While first-generation NNRTIs (nevirapine and efavirenz) have been related largely to hepatic toxicity [15], those belonging to the second generation (etravirine, rilpivirine, and doravirine) seem generally safe for the liver, even in people coinfected with hepatitis viruses [14,16,17,18].

More recently, it has been suggested that, in scenarios that are unrelated to HIV infection, RPV is hepatoprotective, as it exerts a role as an anti-inflammatory, anti-steatotic, and anti-fibrotic agent in different animal models of chronic liver injury [19].

Until now, few data have been available regarding DOR. Serum aminotransferase elevations have been reported in 13% of PLWH on regimens containing this drug, but elevations above five times the upper limit of normal are uncommon, occurring in 1% or less of people. DOR has not been linked to cases of acute hepatitis, acute liver failure, chronic hepatitis, or vanishing bile duct syndrome [20].

Interestingly, in both cohorts, we found a slight ALT increase in PLWH with normal baseline levels and a steep decline in those with elevated baseline ALT treated with DOR. The latter finding derived from a too-small sample (28 study participants) to be significant and needs to be cautiously interpreted. In the RPV cohort, we did not observe significant changes, and grade 3 ALT elevation occurred in three people with concurrent liver disease, none of whom had normal levels at baseline. Noteworthy, comparing HBsAg- or HCV-positive persons with HBsAg- or HCV-negative ones, transaminase changes seemed independent from serostatus. Therefore, our observation confirms the relative liver safety of both RPV and DOR cohorts.

Overall, regimens based on both DOR and RPV were well tolerated; they were similar in terms of interruptions during the first year of observation—the discontinuations due to AEs were infrequent and not significantly different between the two cohorts, but remained unaltered in the DOR cohort. In detail, we recorded 7 and 10 treatment interruptions due to AEs in DOR and RPV cohorts, respectively. The central nervous system (CNS) was involved in 3 and 5 events, showing a low incidence, comparable between cohorts. Undoubtedly, CNS toxicity is a critical issue, being, of the six currently marketed classes of antiretroviral drugs, the NNRTIs most commonly associated with neuropsychiatric complications [21].

In a real-world setting comparing a RPV- to an EFV-containing regimen, discontinuations due to intolerance/toxicity were reported for 15% of RPV- vs. 30% of EFV-treated participants. The main difference was for toxicity of CNS (3% vs. 22%, *p* < 0.001) [22].

In a 96-week phase IIb trial comparing DOR with TDF/FTC safety and efficacy with EFV with TDF/FTC, the first regimen demonstrated superior CNS safety [23].

However, in PLWH who had CNS complaints while receiving EFV/FTC/TDF, improvement in CNS toxicities attributable to EFV was not significantly different after switching to DOR/3TC/TDF compared with remaining on EFV/FTC/TDF [24].

Our real-life observation confirms that discontinuation due to CNS toxicity is a rare occurrence in both groups of treatment. Nonetheless, during the first year of treatment, PLWH in the RPV cohort more frequently interrupted their regimen. Whereas the interruptions due to AEs were similar, the difference appeared due to a greater proportion of therapeutic failures (one in the DOR cohort and six in the RPV cohort) and other reasons (i.e., three participants had to stop RPV because of potential interactions with drugs they were starting for other diseases).

This study had some limitations. First, the Infectious Diseases Clinics involved in the SCOLTA study were not formally representative of Italian Clinics (i.e., at the national level), because they were not randomly selected but participated in our observational study on a voluntary basis. Second, the study participants were not fully representative of all PLWH followed in the Infectious Diseases Clinics participating in the SCOLTA study, but only of those in need of initiating a new ART drug in the considered periods. For these reasons, we cannot fully exclude the potential for unmeasured confounding and channeling bias. Another potential bias may be represented by the different years in which PLWH were enrolled in the two cohorts; although they do not reflect different ART eras and knowledge, previous ART regimens in the RPV group were mainly PIs and TDF-based, while they were mainly INI-based in the DOR group. As regards the analysis of hepatic virus coinfection, a limitation was the lack of information about the duration of HBsAg and HCV coinfections. Lastly, PLWH had different baseline characteristics, reflecting the 7 years difference in enrollment. PLWH in the DOR group were older, had a higher weight and BMI, and a more frequent intake of lipid-lowering drugs. Finally, IDU as a risk factor for HIV acquisition was less represented, although we carefully adjusted for confounding factors. Despite these limitations, our study had the strength to describe a fairly large, real-life cohort of PLWH on DOR-based regimens, followed up prospectively in multiple centers across Italy, in a research network (SCOLTA) specifically designed to improve post-marketing surveillance of adverse reactions to antiretrovirals, and with precise expertise in AE monitoring.

## 5. Conclusions

In conclusion, lipid profile improved in both cohorts, more markedly in people on DOR-based regimens, but with no statistical significance. Moreover, the DOR group showed a significant reduction in ALT levels in PLWH with higher than the normal values at baseline. This finding, though founded on a small sample, seems worthy of further exploration, given the scarcity of information available on the issue.

## Figures and Tables

**Figure 1 viruses-15-01612-f001:**
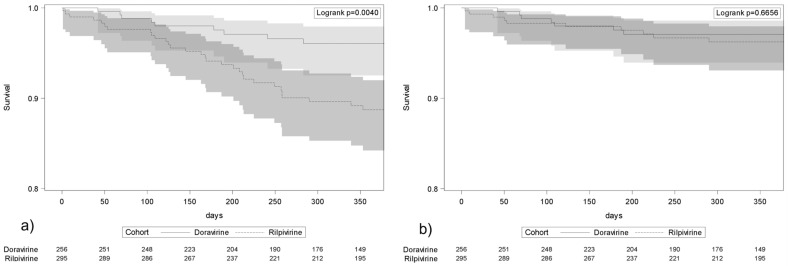
Survival analysis in strata of DOR and RPV regimens. (**a**) Treatment interruptions for any reason and (**b**) treatment interruptions for adverse events. Numbers show the study participants at risk in the doravirine and rilpivirine cohorts.

**Table 1 viruses-15-01612-t001:** Baseline characteristics of 551 ART-experienced patients enrolled in the SCOLTA Project.

Variables at Enrollment	Doravirine (n = 256) Mean ± SD or n (%) or Median (IQR)	Rilpivirine (n = 295) Mean ± SD or n (%) or Median (IQR)	*p*-Value
Age, years (N = 551)	51.9 ± 11.4	46.7 ± 9.4	<0.0001
Male sex (N = 551)	183 (71.5%)	218 (73.9%)	0.52
BMI, kg/m^2^ (N = 489)	26.2 ± 5.3	24.3 ± 3.8	<0.0001
Weight, kg (N = 542)	76.7 ± 15.6	71.9 ± 14.4	0.0002
Caucasian (N = 551)	226 (88.3%)	272 (92.2%)	0.12
Risk factor for HIV acquisition (N = 551)			
Sexual	178 (69.5%)	221 (74.9%)	
IDU	37 (14.4%)	57 (19.3%)	0.0003
Other/ND	41 (16.0%)	17 (5.8%)	
HBsAg-positive (N = 511)	12 (5.3%)	19 (6.6%)	0.54
HCVAb-positive (N = 524)	44 (19.0%)	66 (22.5%)	0.33
Detectable HCV-RNA	6 (11.1%)	44 (55.0%)	<0.0001
Previous ART regimen (N = 551)			
PI	64 (25.0%)	139 (47.1%)	<0.0001
INI	126 (49.2%)	16 (5.4%)	<0.0001
NNRTI	107 (41.8%)	136 (46.1%)	0.31
TDF	49 (19.1%)	235 (79.7%)	<0.0001
Current TDF (N = 551)	146 (57.0%)	271 (91.9%)	<0.0001
CD4, cells/mm^3^ (N = 550)	707 (500–977)	654 (490–842)	0.009
CDC stage C (N = 547)	57 (22.6%)	58 (19.7%)	0.11
Total cholesterol, mg/dL (N = 540)	197 ± 45	191 ± 43	0.13
HDL-cholesterol, mg/dL (N = 539)	49 ± 14	48 ± 17	0.44
Total cholesterol/HDL-cholesterol (N = 538)	4.1 (3.3–4.8)	4.1 (3.2–5.0)	0.75
LDL-cholesterol, mg/dL (N = 537)	118 ± 38	113 ± 38	0.16
Triglycerides, mg/dL (N = 541)	121 (89–175)	120 (85–174)	0.43
On lipid-lowering drugs (N = 551)	57 (22.3%)	28 (9.5%)	<0.0001
Diabetes (N = 551)	13 (5.1%)	13 (4.4%)	0.71
Blood glucose, mg/dL (non-diabetic) (N = 510)	90 ± 16	90 ± 12	0.6
AST, UI/L (N = 524)	22 (18–28)	24 (19–32)	0.004
AST > 40 UI/L	13 (5.1%)	44 (14.9%)	0.0002
ALT, UI/L (N = 528)	22 (17–31)	29 (21–42)	<0.0001
ALT > 40 UI/L	28 (10.9%)	78 (26.4%)	<0.0001

ALT: alanine aminotransferase; ART: antiretroviral therapy; AST: aspartate aminotransferase; HBV: hepatitis B virus; HCV: hepatitis C virus; HDL: high-density lipoprotein; IDU: intravenous drug use; INI: integrase inhibitors; IQR: interquartile range; LDL: low-density lipoprotein; NNRTI: non-nucleoside reverse transcriptase inhibitor; PI: protease inhibitor; SD: standard deviation.

**Table 2 viruses-15-01612-t002:** Change from baseline to T1 (mean, 95% confidence interval) comparison between DOR-based and RPV-based regimens.

T1–T0	Doravirine (n = 256)	Rilpivirine (n = 295)	*p*-Value *
BMI, kg/m^2^	−0.1 (−0.3, 0.1)	**0.1 (0.0, 0.2)**	0.18
Weight, kg	−0.1 (−0.6, 0.4)	**0.3 (0.0, 0.6)**	0.24
Total cholesterol, mg/dL	**−20 (−24, −15)**	**−16 (−20, −12)**	0.89
HDL-cholesterol, mg/dL	−1 (−2, 0)	**−3 (−4, −2)**	0.09
Total cholesterol/HDL-cholesterol	**−0.36 (−0.48, −0.23)**	−0.08 (−0.25, 0.08)	0.39
LDL-cholesterol, mg/dL	**−14 (−19, −10)**	**−9 (−13, −6)**	0.26
Triglycerides, mg/dL	**−22 (−34, −9)**	**−22 (−33, −11)**	0.12
**Not on lipid-lowering drugs**			
Total cholesterol, mg/dL	**−19 (−23, −14)**	**−17 (−21, −13)**	0.34
HDL-cholesterol, mg/dL	−1 (−2, 1)	**−3 (−4, −2)**	0.0005
Total cholesterol/HDL-cholesterol	**−0.33 (−0.45, −0.22)**	−0.05 (−0.22, 0.13)	0.24
LDL-cholesterol, mg/dL	**−14 (−19, −10)**	**−10 (−13, −6)**	0.43
Triglycerides, mg/dL	**−21 (−35, −6)**	**−20 (−32, −8)**	0.14
**Liver enzymes**			***p* ****
AST ≤ 40 UI/L at T0	**2 (1, 3)**	**3 (2, 4)**	0.02
AST > 40 UI/L at T0	−6 (−21, 7)	−1 (−11, 9)	0.44
ALT ≤ 40 UI/L at T0	**3 (2, 5)**	**8 (6, 11)**	0.0001
ALT > 40 UI/L at T0	**−14 (−24, −4)**	−2 (−12, 8)	0.45

ALT: alanine aminotransferase; AST: aspartate aminotransferase; DOR1: lamivudine/tenofovir disoproxil fumarate; DOR2: other DOR-based regimens; HDL: high density lipoprotein; LDL: low density lipoprotein; bold: change from baseline *p* < 0.05 (paired *t*-test); * multivariate analysis of variance, including baseline age, weight, previous use of PI and INI, and previous and current use of TAF and TDF, plus lipid-lowering drug use in the overall sample; ** multivariate analysis of variance, including baseline age, weight, previous and current use of TDF, and hepatic virus coinfections.

**Table 3 viruses-15-01612-t003:** Change from baseline to T1 (mean, 95% confidence interval), comparison between 3TC/TDF/DOR (DOR1) and other DOR-based regimens (DOR2).

T1–T0	DOR1 (n = 142)	DOR2 (n = 114)	*p*-Value *
BMI, kg/m^2^	−0.0 (−0.26, 0.25)	−0.10 (−0.44, 0.23)	0.49
Weight, kg	0.1 (−0.6, 0.7)	−0.1 (−0.9, 0.6)	0.61
Total cholesterol, mg/dL	**−24 (−30, −17)**	**−14 (−21, −7)**	0.53
HDL-cholesterol, mg/dL	−1 (−3, 1)	−1 (−3, 1)	0.35
Total cholesterol/HDL-cholesterol	**−0.44 (−0.61, −0.27)**	**−0.25 (−0.44, −0.06)**	0.36
LDL-cholesterol, mg/dL	**−19 (−23, −13)**	**−9 (−16, −2)**	0.62
Triglycerides, mg/dL	**−19 (−32, −6)**	**−25 (−47, −2)**	0.67
**Not on lipid-lowering drugs**			
Total cholesterol, mg/dL	**−22 (−28, −16)**	**−14 (−22, −7)**	0.52
HDL-cholesterol, mg/dL	−1 (−3, 1)	−0 (−3, 2)	0.41
Total cholesterol/HDL-cholesterol	**−0.37 (−0.51, −0.22)**	**−0.28 (−0.49, −0.07)**	0.25
LDL-cholesterol, mg/dL	**−17 (−23, −12)**	**−10 (−17, −2)**	0.60
Triglycerides, mg/dL	**−15 (−29, −2)**	−29 (−60, 1)	0.78
**Liver enzymes**			***p*-Value ****
AST ≤ 40 UI/L at T0	**2 (0, 4)**	**2 (1, 3)**	0.66
AST > 40 UI/L at T0	−5 (−14, 4)	−1 (−11, 10)	0.72
ALT ≤ 40 UI/L at T0	**4 (2, 6)**	**2 (1, 4)**	0.34
ALT > 40 UI/L at T0	**−20 (−30, −9)**	−7 (−24, 9)	0.38

ALT: alanine aminotransferase; AST: aspartate aminotransferase; DOR1: lamivudine/tenofovir disoproxil fumarate; DOR2: other DOR-based regimens; HDL: high density lipoprotein; LDL: low density lipoprotein; bold: change from baseline *p* < 0.05 (paired *t*-test); * multivariate analysis of variance, including baseline age, weight, previous use of PI and INI, and previous and current use of TAF and TDF, plus lipid-lowering drug use in the overall sample; ** multivariate analysis of variance, including baseline age, weight, previous and current use of TAF and TDF, and hepatic virus coinfections.

**Table 4 viruses-15-01612-t004:** Baseline values (median, IQR) and change from baseline to T1 (mean, 95% confidence interval), comparison between DOR-based and RPV-based regimens in strata of HBsAg and/or HCV positivity, and direct comparison between HBsAg/HCV-positive and -negative persons.

Liver Enzymes, IU/mL	Doravirine (n = 178)	Rilpivirine (n = 213)	Univariate *p*-Value
**HCV/HBsAg-Negative**			
AST T0, median (IQR)	22 (18–28)	23 (18–28)	0.80
AST T1−T0, mean (95% CI)	1 (−0, 2)	**3 (1, 5)**	0.05
ALT T0, median (IQR)	23 (17–31)	27 (20–38)	0.002
ALT T1−T0, mean (95% CI)	1 (−1, 3)	**7 (4, 10)**	0.001
**HCV/HBsAg-Positive**	**Doravirine (n = 54)**	**Rilpivirine (n = 80)**	
AST T0, median (IQR)	22 (17–27)	31 (23–56)	<0.0001
AST T1−T0, mean (95% CI)	2 (−1, 4)	1 (−4, 6)	0.78
ALT T0, median (IQR)	22 (16–30)	42 (28–72)	<0.0001
ALT T1−T0, mean (95% CI)	2 (−2, 6)	3 (−5, 11)	0.82
**Total Sample**	**HCV/HBsAg-Negative (n = 391)**	**HCV/HBsAg-Positive (n = 134)**	
AST T0, median (IQR)	22 (18–28)	27 (20–42)	<0.0001
AST T1−T0, mean (95% CI)	**2 (1, 3)**	1 (−2, 4)	0.54
ALT T0, median (IQR)	25 (18–35)	31 (21–55)	<0.0001
ALT T1−T0, mean (95% CI)	4 (2, 6)	3 (−3, 8)	0.61

**Table 5 viruses-15-01612-t005:** Reasons for interruption during the first year of observation.

Reason	Doravirine N = 256	Rilpivirine N = 295
Any	9 (3.5%)	30 (10.2%)
Adverse event	7 (77.8%)	10 (33.3%)
Arthralgia/asthenia	3	2
Gastrointestinal	1	0
Central nervous system	3	5
Hepatic	0	1
Renal	0	1
Other	0	1
Therapeutic failure	1 (11.1%)	6 (20.0%)
Poor adherence/patient’s will	0	2 (6.7%)
Death *	0	1 (3.3%)
Other	0	6 (20.0%) **
Lost to follow-up	1 (11.1%)	5 (16.7%)

* RPV: heart valve rupture; ** simplification (2), drug–drug interaction (3), not specified (1).

## Data Availability

The data that support the findings of this study are available from the corresponding author, ER, upon reasonable request.

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
