# Peer review of "Lipids and Transaminase in Antiretroviral-Treatment-Experienced People Living with HIV, Switching to a Doravirine-Based vs. a Rilpivirine-Based Regimen: Data from a Real-Life Setting"

_viruses, 2023, doi:10.3390/v15071612_

Round 1

Reviewer 1 Report

Dear Editor,

I carefully read the manuscript "Lipids and transaminase in antiretroviral treatment experienced people living with HIV, switching to a doravirine-based vs rilpivirine-based regimen. Data from a real-life setting".

My comments and suggestions for the authors are the following:

 - Table 1: For each parameter, the authors should specify the sample size including a "N=" in each raw of the table.

 - Tables: In the title, "P" should be replaced by "P-value".

 - The authors should specify how the normal distribution of the variables was assessed.

 - Did the authors perform the Levene's test before the Student's test?

 - Study's limitations should be further and more deeply discussed.

 - English language needs to be carefully revised and improved.

 - The authors should highly consider to refer to doi: 10.33393/dti.2023.2529, doi: 10.1016/j.cpcardiol.2023.101783, doi: 10.3390/jcm12113644, doi: 10.3390/v15051046, doi: 10.1016/j.atherosclerosis.2022.06.001 and doi: 10.1371/journal.pone.0285926.

See my comments below.

Reviewer 2 Report

Dear authors,

thank you for the submission of the manuscript to Viruses. The manuscript is well written, its subject is of high significance. The number of patients that were switched to doravirine (DOR) regimen (256 patients) is comparable with rilpivirine-based regimen (RPV) cohort (295).   However I have some comments that should be taken into account.

Major comments

First, I suppose that the patients having HCVAb positive and HBsAg positive results of testing should be discussed more thoroughly. In particular, what is the duration of hepatitis B (HBV) and hepatitis C (HCV) infection for those patients.

Second, I insist that the group of patients with HBV and HCV should be  considered and analyzed separately. If I understand correctly, the proportion of HBV + HCV positive group was over 24% in the cohort of patients that took doravirine and it was over 29% in rilpivirine cohort. In this case the  comparison of the laboratory tests results measuring hepatic function is not significant. The HBV + HCV positive patients having  DOR regimen should be compared with the same group (HBV + HCV positive) having RPV regimen. And these group additionaly should be compared with the group of HBV and HCV negative people on both DOR and RPV regimens. 

Similarly, 70% of male can be analyzed separately and the results can be provided.

Minor comment:

Please, add the description of the "missing" term in Table  1.

Some spelling errors and typos need to be corrected. 

Reviewer 3 Report

The authors present a manuscript that focuses on metabolic and hepatic profile of PLWH that are switched to a DOR-based therapy. This study compares DOR and the NNRTI RPV, which are now commonly used in ARV. The lipid profile in both drug cohorts are favorable with DOR only showing marked improvements in PLWH that featured slightly elevanted alanine aminotransferase ALT levels at baseline. This seems like a small sample size that might benefit from a study with a larger sample size. Either way, the data is sound, and the writing demonstrates the significance of the data.  

Round 2

Reviewer 2 Report

Dear authors, thank for careful revision of the manuscript. I suppose that it now can be accepted for publication.